# Rodent-Borne Parasites and Human Disease: A Growing Public Health Concern

**DOI:** 10.3390/ani15182681

**Published:** 2025-09-13

**Authors:** Alfonso J. Rodriguez-Morales, Awad A. Shehata, Rokshana Parvin, Shadia Tasnim, Phelipe Magalhães Duarte, Shereen Basiouni

**Affiliations:** 1Faculty of Health Sciences, Universidad Científica del Sur, Lima 15307, Peru; 2Grupo de Investigación Biomedicina, Faculty of Medicine, Fundación Universitaria Autónoma de las Américas-Institución, Universitaria Visión de las Américas, Pereira 660003, Colombia; 3TUM School of Natural Sciences, Bavarian NMR Center (BNMRZ), Structural Membrane Biochemistry, Technical University of Munich, 85748 Garching, Germany; awad.shehata@tum.de; 4Department of Pathology, Faculty of Veterinary Science, Bangladesh Agricultural University, Mymensingh 2200, Bangladesh; rokshana.parvin@bau.edu.bd (R.P.); sadiatasnim562@gmail.com (S.T.); 5Postgraduate Program in Animal Bioscience, Federal Rural University of Pernambuco (UFRPE), Recife 52171-900, Pernambuco, Brazil; duarte.phe@gmail.com; 6Center for Preclinical Research, German Cancer Research Center, 69120 Heidelberg, Germany

**Keywords:** rodents, zoonotic parasites, public health, protozoa, helminths, ectoparasites, one health

## Abstract

Rodents are among the most widespread mammals globally and serve as key reservoirs for many parasites that can infect humans. These parasites include protozoa, helminths, and ectoparasites, which are capable of causing serious diseases such as toxoplasmosis, Chagas disease, leishmaniasis, and trichinellosis. The spread of these parasites is exacerbated by factors such as urbanization, inadequate sanitation, and climate change, which increase the likelihood of contact between rodents and humans. This review highlights the growing public health threat of rodent-borne parasites. It emphasizes the importance of One Health approaches that integrate environmental management, rodent control, public education, and global surveillance. Understanding and addressing these risks is essential to protect both human and animal health.

## 1. Introduction

Rodents are among the most widespread and adaptable mammals on Earth, inhabiting a diverse range of ecosystems, from rural environments to densely populated urban centers. Rodentia is an order within the class Mammalia, subclass Theria (live-bearing mammals), and infraclass Eutheria (placental mammals) (Figure 1) (Table 1). It belongs to the superorder Euarchontoglires, which also includes primates, lagomorphs (such as rabbits and hares), and treeshrews. Within Rodentia, over 2000 recognized species are grouped into several suborders (Figure 1), including Myomorpha (which comprises the family Muridae, including true rats, true mice, and gerbils) (Figure 2), Sciuromorpha (squirrels and chipmunks), Hystricomorpha (porcupines and guinea pigs), and others. This order represents the largest and most diverse group of mammals, accounting for approximately 40% of all known mammalian species. Rodents are unified by a distinct dental formula, most notably a single pair of continuously growing incisors in both the upper and lower jaws, which they use for gnawing and feeding [1,2,3].

Their proximity to human habitats, high reproductive rates, and capacity to carry a wide array of pathogens make them key reservoirs for zoonotic diseases. As urbanization, environmental disruption, and climate change continue to reshape rodent populations and their interactions with humans, the risk of transmission of rodent-borne pathogens is increasing. These dynamics have significant implications for public health, particularly in resource-limited settings where surveillance and control measures may be inadequate. Understanding the role of rodents in the ecology of infectious diseases is essential for developing effective prevention and mitigation strategies [4,5,6].

Rodents are hosts to numerous zoonotic parasites that pose significant public health threats. Among the protozoan parasites, *Toxoplasma gondii* is particularly concerning, as it causes toxoplasmosis, which can have a severe impact on immunocompromised individuals and pregnant women. *Trypanosoma cruzi*, responsible for Chagas disease, is transmitted through triatomine bugs and can lead to severe cardiac and digestive complications. Other protozoa, such as *Giardia intestinalis* and *Cryptosporidium* spp., also infect rodents and can be transmitted to humans through contaminated water, leading to gastrointestinal and systemic diseases [7,8].

Several helminths found in rodents also pose zoonotic risks. *Hymenolepis diminuta* and *Hymenolepis nana* are tapeworms that can infect humans via contaminated food or water. *Trichinella spiralis* causes trichinellosis, a severe disease associated with the consumption of undercooked, infected meat. *Angiostrongylus cantonensis*, known as the rat lungworm, is responsible for eosinophilic meningitis in humans, while *Capillaria hepatica* can cause hepatic disease. Additionally, *Baylisascaris procyonis*, a nematode found in rodents, has been associated with severe neurological disorders in accidental human hosts. *Echinococcus multilocularis*, the causative agent of alveolar echinococcosis, involves rodents as intermediate hosts and can lead to life-threatening hepatic and systemic disease in humans. Beyond helminths, rodent-associated protozoa such as *Babesia microti*, the primary agent of human babesiosis, and *Entamoeba histolytica*, responsible for amoebiasis, further highlight the significant zoonotic potential of these animals [9,10,11].

Ectoparasites that infest rodents also contribute to the spread of zoonotic diseases. The oriental rat flea (*Xenopsylla cheopis*) is the primary vector of *Yersinia pestis*, the causative agent of plague, while *Rickettsia typhi*, transmitted by fleas, leads to murine typhus. The tropical rat mite (*Ornithonyssus bacoti*) can cause dermatitis in humans, which may lead to secondary infections. Various tick species (*Ixodes* spp.) and lice can also transmit diseases such as Lyme disease and relapsing fever, respectively [12,13,14].

The presence of these zoonotic parasites in rodents highlights the importance of vector control and public health interventions. Poor sanitation, urbanization, and inadequate waste management contribute to the growth of rodent populations, thereby increasing the risk of disease transmission. Surveillance, rodent control programs, and hygiene measures are crucial in preventing outbreaks and mitigating the impact of rodent-associated parasitic infections on human health (Table 2) [15,16].

Rodents play diverse and critical roles in the transmission of zoonotic pathogens, and their classification as hosts depends on the ecological and biological context of each pathogen. As intermediate hosts, rodents may harbor larval or developmental stages of parasites before transmission to a definitive host. As reservoir hosts, they maintain the pathogen in nature, ensuring its persistence even in the absence of human cases. In some instances, rodents serve as definitive hosts, supporting the sexual reproduction of the parasite [17,18]. The concept of a competent reservoir is particularly relevant, as particular rodent species not only maintain the pathogen but also transmit it efficiently to vectors or directly to humans, thus sustaining outbreaks. Finally, when rodents are considered essential reservoirs, their presence is indispensable for the persistence of the pathogen in a given ecosystem. Clarifying these distinctions is crucial for both parasitologists and non-specialists, as it allows a more accurate understanding of the epidemiological significance of rodents in zoonotic transmission cycles and informs targeted interventions under the One Health framework [19,20].

## 2. Protozoa

### 2.1. Toxoplasma gondii

*Toxoplasma gondii* is a highly significant protozoan parasite that infects rodents and plays a crucial role in its life cycle (Figure 3). Rodents serve as intermediate hosts, harbouring the parasite in tissue cysts, particularly in the brain and muscles. This infection can alter rodent behaviour, making them more susceptible to predation by felines, the definitive hosts. This behavioural alteration is believed to result from the parasite’s interference with neural signalling in the amygdala, a small, almond-shaped brain structure located deep within the temporal lobes. The amygdala is central to the processing of fear, anxiety, aggression, memory, and decision-making. When its normal function is disrupted, rodents may fail to respond appropriately to threats, increasing their risk of predation by cats. This manipulation enhances the likelihood of the parasite completing its life cycle (Table 2), representing a remarkable example of evolutionary adaptation. *Toxoplasma* is known to modify not only the behavior of its intermediate animal hosts but also the behavior and personality of infected humans [21].

A meta-analysis identified that the overall seroprevalence was calculated at 6% (95%CI = 6–7%), with the highest amount observed in Africa (24%) and South America (18%) [22]. Research conducted in Egypt found that 18% (44 out of 244) of examined rodents tested positive for *T. gondii* DNA using PCR methods [23]. *Toxoplasma gondii* was first described in a rodent, *Ctenodactylus gundi* [24]. A study in Croatia and Slovenia detected *T. gondii* DNA in 2.94% of rodents across five species, including *Rattus rattus* and *Mus musculus*, collected from waste disposal sites [25].

From a public health perspective, rodents play a critical role in environmental contamination. Although they are not definitive hosts and do not shed oocysts directly, their presence in human habitats contributes to the maintenance of *T. gondii* transmission cycles. They may serve as prey for cats, which then shed infectious oocysts in their faeces, contaminating soil, water, and food sources. Consequently, controlling rodent populations and improving hygiene practices are essential measures for reducing human exposure and preventing toxoplasmosis [26,27].

### 2.2. Trypanosoma cruzi

*Trypanosoma* species are protozoan parasites with significant implications for both animal and human health [28,29] (Table 2). Rodents serve as natural reservoirs for various *Trypanosoma* species, facilitating their maintenance and transmission within ecosystems (Figure 4). *Trypanosoma cruzi*, the causative agent of Chagas disease, utilizes rodents as reservoir hosts, playing a crucial role in the epidemiology of this disease. Given the global distribution of these parasites, monitoring rodent populations is essential for understanding transmission dynamics and mitigating public health risks [30,31].

Several studies have investigated the prevalence of *Trypanosoma* infections in rodent populations. A study conducted in San Luis province, Argentina, examined wild rodents for *T. cruzi* infection. Out of 25 isolates obtained, 44% were identified as *T. cruzi*, while the remaining 56% were classified as *T. cruzi*-like organisms. This research highlights the role of wild rodents in maintaining the transmission cycle of *T. cruzi* in areas with low endemicity [32]. Research in a rural community in Chile assessed *T. cruzi* infection in synanthropic and wild rodents. The study found that 83.1% of the rodents were infected with *T. cruzi*, with parasite loads comparable to those observed in human cases. This high prevalence underscores the potential risk of zoonotic transmission in rural settings [33]. Meanwhile, an analysis in New Mexico, USA, identified an 11% infection rate of *T. cruzi* among 1428 rodents tested, underscoring the role of rodents in the transmission dynamics of Chagas disease [34]. It should be noted here that the morphological and serological methodology of many studies did not allow for differentiation between *T. cruzi* and the non-pathogenic *T. rangeli*, so these data on the prevalence of *T. cruzi* in rodents may be overestimated.

These studies highlight the importance of rodent surveillance in controlling *Trypanosoma* infections. Rodents contribute to the environmental spread of these parasites, which can potentially lead to zoonotic outbreaks. Understanding their prevalence in different regions helps implement targeted vector control measures, reducing the risk of human and animal infections [35].

### 2.3. Leishmania spp.

*Leishmania* species are protozoan parasites responsible for leishmaniasis, a disease affecting humans and animals [36,37,38,39,40,41,42] (Table 2), still highly prevalent worldwide and concerning, especially the visceral form associated with significant fatal outcomes (Table 3). Rodents play a crucial role as reservoir hosts in the transmission cycle of various *Leishmania* species (Figure 5). Their close association with human habitats facilitates the spread of the parasite to sand fly vectors, which subsequently transmit the infection to humans (Figure 5) and other animals, especially dogs for *Leishmania infantum/chagasi* (Figure 6). Understanding the prevalence of *Leishmania* in rodent populations is essential for developing effective control strategies and reducing the incidence of leishmaniasis [43,44].

Several studies have investigated the prevalence of *Leishmania* infections in rodents. Research in Barcelona, Spain, revealed a 33.3% prevalence of *Leishmania* infection among Norway rats (*Rattus norvegicus*) inhabiting the city’s sewer system [45]. This high infection rate suggests that urban rodent populations can serve as significant reservoirs for the parasite, potentially impacting public health. However, despite the high prevalence, this is not sufficient evidence to incriminate a reservoir host of *Leishmania* (https://www.who.int/publications/i/item/WHO-TRS-949, accessed on 1 August 2025) [46]. Xenodiagnostic experiments are needed to confirm the role of the suspected host species in *Leishmania* transmission. There are examples where xenodiagnoses have confirmed the reservoir role of the rodent species: *Leishmania infantum*—*Rattus rattus* [47], *L. panamensis—Proechimys semispinosus* [48], *L. braziliensis—Necromys lasiurus*, *Nectomys squamipes*, and *Rattus rattus* [49], *L. major—Meriones shawi* [50].

A study conducted in Greece found that 54.55% of examined rodents had been exposed to *Leishmania* spp., as evidenced by serological tests [51]. This indicates a substantial exposure rate among rodent populations, underscoring their role in the epidemiology of leishmaniasis in the region. In northeastern Brazil, a longitudinal study reported persistent *Leishmania (Viannia) braziliensis* infections in wild rodent populations over 13 months. Notably, a median of 48% of exposed sandflies became infected after feeding on these rodents, highlighting the rodents’ significant role in maintaining the transmission cycle of leishmaniasis [52].

These studies emphasize the importance of monitoring rodent populations as part of integrated leishmaniasis control programs [45,51,52]. Public health initiatives can more effectively reduce the transmission of *Leishmania* to humans by identifying and targeting rodent reservoirs. Surveillance, vector control, and habitat management are key strategies in limiting the spread of this parasitic disease.

### 2.4. Giardia intestinalis

*Giardia intestinalis* (also known as *G. duodenalis* or *G. lamblia*) is a flagellated protozoan parasite that causes giardiasis, an enteric disease characterized by diarrhea, malabsorption [53], and abdominal discomfort, which is still prevalent, especially in low and middle-income countries [54,55]. Rodents are significant reservoirs for *Giardia*, and multiple genotypes (assemblages) have been identified in these hosts, some of which are zoonotic [56]. The parasite is transmitted via the fecal-oral route, primarily through the ingestion of cysts in contaminated water, food, or environments (Table 2) [57].

Studies have demonstrated a high prevalence of *G. intestinalis* in wild and synanthropic rodent populations [58,59]. In urban areas, rodents such as *Rattus norvegicus* and *Mus musculus* frequently harbor *Giardia* cysts, contributing to environmental contamination [60,61]. In agricultural and peri-urban settings, wild rodent species have also been implicated in maintaining *Giardia* in the environment, potentially impacting livestock and human populations [61]. In a study in Qatar, *Giardia* spp. were detected in 4.1% of rodents by microscopy, predominantly in young animals. However, PCR confirmed that these were not *Giardia intestinalis*, suggesting the presence of non-zoonotic species, such as *G. muris* or *G. microti* [62].

Molecular studies have revealed that rodents carry both host-specific and zoonotic *Giardia* assemblages, including Assemblages A and B, which are commonly found in humans. This suggests the potential for direct or indirect zoonotic transmission from rodents to humans, particularly in settings with poor sanitation or inadequate water treatment. Additionally, rodents may act as sentinels for environmental contamination with *Giardia*, underscoring their importance in public health surveillance [61,63].

A study in Hungary found that by parasitological examination, cysts in 58.3% of asymptomatic Norway rats and 27.6% of chinchillas were identified [58]. Additionally, *Giardia* infection was detected in three degus (prevalence: 16.7%) using the flotation technique. PCR analysis targeting three genetic markers yielded a positivity rate of 3.2%, while flotation revealed a higher prevalence of 21.9%. DNA sequencing was successfully performed on PCR products from five samples. Phylogenetic analysis of partial beta-giardin gene sequences indicated the presence of assemblages B and G in rats [58].

In a systematic review and meta-analysis of five million animals, *G. intestinalis* was detected in 19.3% of rodents, with assemblages A and B identified, indicating potential zoonotic transmission from wild rodents to humans in the study area [64].

The presence of *Giardia* in rodent populations underscores the need for integrated monitoring approaches. Improvements in water, sanitation, waste disposal, and rodent control are essential strategies to reduce the risk of giardiasis outbreaks in human communities. Surveillance programs that include genotyping of *Giardia* isolates from rodents, humans, and water sources can help elucidate transmission pathways and inform effective public health interventions [65,66,67].

### 2.5. Cryptosporidium spp.

*Cryptosporidium* spp. are apicomplexan protozoan parasites that cause cryptosporidiosis [68,69], an intestinal disease marked by watery diarrhea, dehydration, and weight loss [70,71]. While several *Cryptosporidium* species infect humans, rodents have emerged as essential reservoirs for both rodent-adapted and zoonotic species, such as *C. parvum*, *C. muris*, and *C. hominis* [72,73]. Transmission occurs through ingestion of oocysts, which are excreted in feces and can contaminate water, food, or fomites (Table 2).

Rodent populations in both urban and rural environments frequently harbor *Cryptosporidium* spp., with varying infection rates depending on species, location, and season. Studies from urban centers in Asia and Latin America have reported *Cryptosporidium* prevalence in rats ranging from 5% to over 25%, with higher rates often associated with inadequate sanitation infrastructure. In agricultural contexts, wild rodents can contaminate water supplies and crops, increasing the risk of transmission to livestock and humans [74,75].

Molecular analyses have revealed that rodents not only carry rodent-specific species (e.g., *C. muris*, *C. andersoni*) but also zoonotic types, particularly *C. parvum*, which poses a significant public health concern. The presence of zoonotic genotypes in rodent feces, particularly in areas with high human population densities, suggests a role for rodents in environmental contamination and potential outbreaks. Oocysts are resistant to conventional chlorination, making their presence in water systems particularly problematic [76,77,78].

A recent systematic review found that *Cryptosporidium* spp. were detected in 8.8% of rodents, with several novel genotypes identified, suggesting wild rodents may serve as reservoirs for diverse and potentially zoonotic *Cryptosporidium* species [79].

Rodents may serve as both amplifiers and indicators of environmental *Cryptosporidium* contamination. In developing regions, cryptosporidiosis is a major contributor to childhood diarrheal disease, and the role of rodents in sustaining transmission cycles should not be underestimated. Control measures focusing on rodent exclusion from food and water sources, combined with improved water treatment and public education, are critical components of prevention strategies [80,81,82].

Given the protozoan’s resilience in the environment and the diversity of *Cryptosporidium* species harbored by rodents, comprehensive surveillance, including genotyping and environmental monitoring, is necessary to understand and mitigate the zoonotic risks associated with rodent populations [83,84].

### 2.6. Babesia microti

*Babesia microti* is an intraerythrocytic protozoan parasite and the primary etiological agent of human babesiosis, an emerging tick-borne zoonosis with global health significance. Rodents, particularly members of the family Cricetidae such as the white-footed mouse (*Peromyscus leucopus*), serve as the principal reservoirs of *B. microti*. These small mammals play a pivotal role in maintaining the enzootic cycle by harboring the parasite in their erythrocytes and enabling transmission to the vector, the *Ixodes* tick [85,86].

Rodents are crucial in sustaining *B. microti* transmission dynamics, as their population density and distribution strongly influence infection prevalence in vector populations. High rodent abundance increases opportunities for ticks to feed on infected hosts, thereby amplifying parasite circulation within endemic areas. Moreover, the long duration of parasitemia in rodent hosts ensures continued availability of the parasite to feeding ticks over extended periods [87,88].

Human babesiosis occurs when infected *Ixodes* ticks bite humans, introducing the parasite into the bloodstream. In humans, the disease can range from asymptomatic infection to severe, malaria-like illness, especially in immunocompromised or splenectomized individuals. Although humans are incidental hosts and not directly infected by rodents, the latter’s role as reservoirs is indispensable for sustaining the parasite’s natural cycle and thus represents a critical link in zoonotic transmission [87,88].

The importance of rodents in the epidemiology of *B. microti* extends beyond ecological maintenance; they also act as sentinel hosts for surveillance programs. Monitoring infection prevalence in rodent populations provides an early warning system for assessing human risk in endemic regions. Therefore, understanding rodent ecology and host–vector interactions is essential for designing effective preventive strategies against babesiosis [88,89].

### 2.7. Entamoeba histolytica

*Entamoeba histolytica* is a protozoan parasite and the causative agent of human amoebiasis, a significant public health concern in many developing regions. Although humans are the primary hosts of *E. histolytica*, rodents can play an essential role in its epidemiology, serving as potential reservoirs and experimental hosts. The parasite primarily colonizes the large intestine, where it can exist in two forms: the infective cyst, which is environmentally resistant, and the trophozoite, which can invade intestinal tissues and cause disease [90,91].

In rodent populations, natural infections with *Entamoeba* species, including *E. histolytica* and closely related commensal species, have been reported. These animals may contribute to environmental contamination with infective cysts through their feces, thereby sustaining transmission cycles in areas with poor sanitation. Furthermore, rodents are frequently used as experimental models to study host–parasite interactions, pathogenesis, and potential therapies for amoebiasis, highlighting their importance not only in natural epidemiology but also in biomedical research [11,92].

The zoonotic significance of *E. histolytica* lies in its potential to cross species barriers under favorable ecological conditions. In humans, infection can range from asymptomatic colonization to invasive disease, including amoebic dysentery and extraintestinal manifestations such as liver abscesses. Rodent populations in close contact with human settlements, particularly in urban slums and agricultural areas, may contribute to maintaining environmental contamination and increasing human exposure risk [93,94].

Overall, while the direct role of rodents as reservoirs of human amoebiasis is still under investigation, their ecological ubiquity, capacity to harbor the parasite, and close association with human environments underscore their importance in the transmission dynamics of *E. histolytica*. Surveillance of rodent populations and improvements in sanitation are, therefore, crucial components of strategies aimed at controlling amoebiasis in endemic areas [95,96].

## 3. Helminths

### 3.1. Hymenolepis spp.

*Hymenolepis* spp. are cestode parasites commonly found in rodents, with two species, *Hymenolepis (H.) diminuta* and *H. nana* (Figure 7), having zoonotic potential [15,97]. These tapeworms are frequently used as indicators of environmental contamination and can pose health risks to humans, particularly in areas with poor sanitation and high rodent activity (Table 2) [98].

*H. diminuta*, the rat tapeworm, primarily infects rodents as definitive hosts and requires arthropods (such as beetles or fleas) as intermediate hosts for its life cycle. Rodents acquire the infection by ingesting infected arthropods, while accidental ingestion of contaminated food can lead to human infection, particularly in children [100,101]. Although *H. diminuta* infections in humans are relatively rare, they are documented in both urban and rural settings, and their presence is considered a marker of poor hygiene and rodent infestation. In rodent populations, prevalence can vary widely, from 5% to over 50%, depending on ecological conditions and the presence of intermediate hosts [102,103].

In a systematic review and meta-analysis, it was found that *H. diminuta* (21.2%) and *H. nana* (13.4%) were prevalent in rodents, indicating significant zoonotic potential and highlighting public health concerns in the studied urban environments [104].

*H. nana*, also known as the dwarf tapeworm, is more epidemiologically significant due to its direct life cycle and ability to autoinfect the host. Unlike *H. diminuta*, *H. nana* does not require an intermediate host, although it can use arthropods facultatively [105,106]. This unique feature enables it to amplify within rodent hosts rapidly—and humans—resulting in high worm burdens and persistent infections. *H. nana* is considered the most common cestode infection in humans globally, particularly in children, and is closely associated with rodent exposure and fecal-oral transmission. Rodents such as *Rattus norvegicus* and *Mus musculus* are important reservoirs, and studies in endemic regions have reported infection rates exceeding 60% in rodent populations [107,108].

Rodents infected with *Hymenolepis* spp. contribute to environmental contamination by shedding eggs in feces, which can persist under favourable conditions. Human infection may lead to symptoms such as abdominal pain, diarrhea, anorexia, and irritability, although many cases remain asymptomatic. The autoinfective nature of *H. nana* poses additional challenges in control, as reinfection can occur without further environmental exposure [100,109].

Public health interventions should prioritize rodent control, improved sanitation, and health education to minimize human exposure to diseases. Regular monitoring of rodent populations for *Hymenolepis* spp. can help identify areas at risk and inform integrated parasite management strategies. Deworming programs in at-risk populations, especially school-aged children, may be necessary in endemic regions with high human-rodent contact [15,110].

### 3.2. Trichinella spiralis

*Trichinella spiralis* is a parasitic nematode responsible for trichinellosis (trichinosis), a zoonotic disease acquired through the consumption of raw or undercooked meat containing infective larvae. Rodents play a key role in the sylvatic and synanthropic transmission cycles of *T. spiralis*, acting as both reservoirs and amplifiers of the parasite (Table 2) [111,112].

Infected rodents harbor encysted larvae in their striated muscles. Transmission occurs when carnivorous or omnivorous animals, including other rodents, consume infected tissue, perpetuating the parasite’s life cycle. Rodent cannibalism, predation, or scavenging behavior facilitates intra- and inter-species transmission. Their interaction with pigs, particularly in unregulated farming systems, creates a key epidemiological interface that should not be underestimated. This ecological dynamic also creates a link between wild rodent populations and domestic or peridomestic animals, such as pigs, which are the principal source of human infections [113,114].

Epidemiological studies have demonstrated that *T. spiralis* can circulate silently in rodent populations across various habitats. In agricultural settings, rodents foraging in or around pig pens and food storage areas increase the likelihood of transmitting infection to swine, especially in backyard or poorly regulated farming systems. Surveys in rural areas of Asia, Eastern Europe (Table 4), and Latin America have reported *T. spiralis* prevalence in wild and synanthropic rodents ranging from 1% to over 10%, depending on trapping location and diagnostic method [115,116]. Although direct transmission of *T. spiralis* from rodents to humans is rare, cultural practices involving the consumption of rodent meat have been implicated in certain outbreaks. More commonly, rodents contribute indirectly by infecting pigs, which humans later consume.

Although human trichinellosis is primarily associated with the consumption of pork, outbreaks have been linked to the ingestion of infected wild game or rodent meat in particular cultural contexts. The role of rodents in maintaining *Trichinella* spp. in the environment highlights the need for continued vigilance, particularly in regions with overlapping wildlife-livestock-human interfaces. Infected rodents can also serve as sentinels for environmental contamination and provide early warning of potential risks to animal and human health [117,118].

Control of *T. spiralis* requires an integrated One Health approach that includes rodent control in pig-rearing facilities, proper cooking of meat, routine veterinary inspection of pork, and public awareness campaigns about the risks of consuming undercooked animal products. Surveillance of rodent populations, especially in endemic areas, can support efforts to interrupt the zoonotic transmission cycle and prevent human trichinellosis outbreaks [119,120].

### 3.3. Angiostrongylus cantonensis

*Angiostrongylus cantonensis*, commonly known as the rat lungworm, is a neurotropic nematode that causes eosinophilic meningitis in humans. Rodents, particularly *Rattus norvegicus* and *Rattus rattus*, serve as the definitive hosts of this parasite, while mollusks such as snails and slugs act as intermediate hosts (Table 2). Rodents acquire infection by consuming infected gastropods, and in turn, they harbor adult worms in the pulmonary arteries. First-stage larvae (L1) are subsequently shed in rodent feces, contaminating the environment and facilitating the parasite’s life cycle [121,122].

The global distribution of *A. cantonensis* has expanded significantly in recent decades, primarily due to the movement of infected rats and gastropods via trade, travel, and climate change-driven ecological shifts. Originally endemic to Southeast Asia and the Pacific Islands, the parasite has now been reported in parts of Africa, the Americas, and the Caribbean. Rodents play a central role in this geographic spread, serving as mobile reservoirs that sustain local transmission cycles [123,124].

Numerous studies have confirmed high prevalence rates of *A. cantonensis* in rodent populations across endemic areas. For instance, surveys in urban and rural sites in Southeast Asia have shown infection rates of 20–60% in *Rattus* species. At the same time, emerging reports in Brazil and the southern United States highlight the parasite’s introduction and establishment in new ecological niches. *A. cantonensis* has been identified in rodents in Spain and other countries of Europe [121,125,126,127,128]. The presence of *A. cantonensis* in synanthropic rodents increases the risk of human exposure, especially in regions with poor sanitation and high gastropod populations [129,130,131].

Human infection occurs through accidental ingestion of infective third-stage larvae (L3), typically by consuming raw or undercooked snails, slugs, freshwater shrimp, or contaminated vegetables. While rodents do not directly transmit the parasite to humans, their presence is crucial for sustaining the life cycle and contributing to environmental contamination with L1 larvae. Human angiostrongyliasis manifests with severe neurological symptoms, including headache, stiff neck, and eosinophilic meningitis, and in some cases may lead to long-term sequelae or death [132,133,134].

Public health efforts to control *A. cantonensis* must include rodent population management, reduction in gastropod intermediate hosts, and education on safe food handling and consumption practices. Integrated surveillance systems that include rodent monitoring can help identify risk areas and facilitate timely interventions. In newly affected regions, early detection in rodent hosts can serve as a sentinel indicator for the parasite’s emergence and guide public health responses [135,136].

### 3.4. Capillaria hepatica

*Capillaria hepatica* (syn. *Calodium hepaticum*) is a zoonotic nematode that infects the liver of mammals, particularly rodents, which serve as the primary reservoir hosts. This parasite causes hepatic capillariasis, a rare but potentially severe disease in humans, characterized by granulomatous hepatitis and hepatic dysfunction. Unlike most helminths, *C. hepatica* requires host death and decomposition or predation for egg dissemination, making its life cycle unique among zoonotic parasites (Table 2) [137,138].

Rodents, especially *Rattus norvegicus* and *Mus musculus*, as well as various wild species, are frequently infected with *C. hepatica* and play a central role in maintaining its sylvatic and synanthropic transmission cycles. The parasite’s eggs are deposited in the liver parenchyma but are only released into the environment following the host’s death, scavenging, or cannibalism by other organisms. Once in the environment, the eggs must embryonate under favorable conditions before becoming infective to new hosts via ingestion [15,139,140].

In rodent populations, prevalence rates can be surprisingly high, with studies from urban and peri-urban environments in South America, Africa, and Asia reporting infection rates ranging from 5% to over 30%, depending on ecological and environmental conditions. The high density and rapid turnover of rodent populations, especially in cities with poor waste management, facilitate the sustained transmission of *C. hepatica* [137,138].

Human infections are rare but likely underreported, as diagnosis requires liver biopsy or histopathological examination, and clinical presentation often mimics other hepatic diseases such as tuberculosis, neoplasms, or viral hepatitis. Documented cases have mainly involved children and individuals with close contact to rodents or poor hygiene conditions. Symptoms may include hepatomegaly, fever, eosinophilia, and elevated liver enzymes, occasionally progressing to liver fibrosis or cirrhosis [141].

The public health significance of *C. hepatica* lies not only in its potential to cause human disease but also in its role as an indicator of environmental rodent infestation and fecal contamination. Control efforts should prioritize rodent management, improved sanitation, and education on minimizing exposure to contaminated soil or rodent carcasses. Monitoring rodent populations in high-risk areas, particularly those in proximity to human dwellings or food storage sites, can support early detection and prevention strategies for hepatic capillariasis [142].

### 3.5. Baylisascaris procyonis

*Baylisascaris procyonis* is a large ascarid nematode primarily parasitizing raccoons (*Procyon lotor*) as definitive hosts, but rodents and other small mammals serve as critical paratenic (transport) hosts. In paratenic hosts such as rodents, the ingested infective eggs hatch and larvae migrate through tissues, particularly the central nervous system (CNS), causing visceral, ocular, and neural larva migrans (Table 2). Though human infections are rare, they are often severe or fatal, making *B. procyonis* a parasite of significant public health concern [143,144].

Rodents, including species such as *Peromyscus* spp., *Microtus* spp., and *Mus musculus*, are frequently involved in the sylvatic life cycle of *B. procyonis*. After ingesting embryonated eggs from raccoon feces-contaminated environments, larvae migrate to the CNS, where they can cause behavioral changes, such as disorientation or ataxia. These neurological impairments make the rodents more susceptible to predation by raccoons, thereby completing the life cycle [145,146].

Although rodents are not the definitive hosts, their role as reservoirs of infective larvae is epidemiologically critical. Infected rodents can harbor hundreds of migrating larvae, creating a substantial risk to predators, including raccoons, domestic animals, and humans. Human infections occur through accidental ingestion of embryonated eggs from contaminated soil, water, or fomites. Children are particularly vulnerable due to geophagia and poor hygiene [147,148].

Environmental contamination with raccoon feces in peri-urban and suburban areas has become increasingly common, especially in North America, where raccoon populations have expanded into human-inhabited environments. Studies have shown that up to 70–80% of raccoons in some regions are infected with *B. procyonis*, and eggs can persist in the environment for years under favorable conditions. This high prevalence among raccoons and the frequent exposure of rodents to raccoon latrines amplify transmission risk [146,149].

Due to the severe neuropathology caused by larval migration, human *Baylisascaris* infections are often devastating. Clinical manifestations include eosinophilic meningoencephalitis, coma, and death. Survivors may experience permanent neurological deficits. No specific antemortem diagnostic test exists, and treatment is limited, especially in late-stage disease [150,151].

Preventive strategies should focus on reducing environmental exposure to raccoon feces, discouraging the feeding of raccoons, and minimizing human contact with contaminated areas. Rodent control also plays an indirect but valuable role by reducing the availability of infected paratenic hosts and limiting parasite propagation in ecosystems. Public health education about the risks of *B. procyonis* is critical in areas where raccoons and humans coexist closely [148,152].

### 3.6. Echinococcus multilocularis

*Echinococcus multilocularis* is a small cestode of the family Taeniidae and the causative agent of alveolar echinococcosis (AE), one of the most severe zoonotic helminthic diseases. The parasite has a complex life cycle involving canids (primarily foxes, but also dogs and other wild carnivores) as definitive hosts, and small mammals, particularly rodents, as intermediate hosts. Rodents play a critical role in maintaining the sylvatic cycle of *E. multilocularis*, serving as the main reservoirs that sustain transmission in natural ecosystems [153,154].

In rodents, larval stages develop as alveolar metacestodes predominantly in the liver, forming proliferative, tumor-like lesions that can be lethal to the host. High prevalence rates in rodent populations, especially in arvicoline rodents such as voles (*Microtus* spp., *Arvicola terrestris*, *Clethrionomys glareolus*), directly influence transmission intensity and the risk of spillover to humans. Humans, as accidental intermediate hosts, acquire the infection through ingestion of parasite eggs excreted by definitive hosts, leading to progressive hepatic lesions with potential metastasis to other organs [155,156].

The importance of rodents in the epidemiology of *E. multilocularis* lies not only in their role as reservoirs but also in their population dynamics, which determine parasite transmission patterns. Fluctuations in rodent abundance, driven by ecological factors such as habitat availability, climate conditions, and predator–prey interactions, strongly impact the prevalence of infection in definitive hosts. Areas with high rodent density and species diversity often show higher levels of environmental contamination with parasite eggs, thereby increasing human exposure risk [157,158].

The public health relevance of *E. multilocularis* is particularly significant in endemic regions of Europe, Asia, and North America, where rodent populations sustain the enzootic cycle. Control strategies must therefore consider the ecological role of rodents, emphasizing surveillance of small mammal communities as sentinels for transmission risk and implementing measures that reduce human contact with infected definitive hosts and contaminated environments [159,160].

## 4. Ectoparasites

### 4.1. Xenopsylla cheopis

*Xenopsylla cheopis*, commonly known as the oriental rat flea, is one of the most medically essential ectoparasites associated with rodents. It is the primary vector of *Yersinia pestis*, the etiologic agent of plague, and also transmits *Rickettsia rickettsii,* the causative agent of Rocky Mountain spotted fever. This flea species predominantly parasitizes commensal rodents such as *Rattus norvegicus* and *Rattus rattus*, which are abundant in urban and peri-urban environments globally (Table 2) [13,14].

Adult *X. cheopis* feed on the blood of rodents, but they may also bite humans when rodent hosts are scarce or when human-rodent contact increases, particularly during outbreaks or in poorly maintained dwellings. The flea becomes infected with *Y. pestis* after feeding on a bacteremic rodent and subsequently transmits the bacterium to new hosts by regurgitating infected blood during subsequent feeding attempts. Blockage of the flea’s proventriculus enhances transmission efficiency [161,162].

Historically, *X. cheopis* played a central role in the global spread of plague during the third pandemic and remains a threat in endemic regions across Africa, Asia, and the Americas. Surveillance studies in these regions consistently report high flea indices (i.e., average number of fleas per rodent) in areas where plague persists. For example, in Madagascar and the Democratic Republic of the Congo, *X. cheopis* is frequently recovered from peridomestic rodents during plague outbreaks [161,162,163].

Besides being a competent vector of plague, *X. cheopis* is also a competent vector of *R. typhi*, which causes murine typhus—a febrile illness characterized by headache, rash, and systemic symptoms. While murine typhus is generally less lethal than plague, its burden in endemic areas is underappreciated, and outbreaks often go undiagnosed due to nonspecific clinical features [164,165].

Environmental factors, including poor sanitation, high rodent densities, and warm climates, favor the proliferation of *X. cheopis*. Its life cycle involves egg deposition in rodent nests and burrows, with larval development occurring in organic debris. Urbanization and unregulated waste management contribute to the persistence of flea populations and increase the risk of human exposure [166,167].

Control strategies against *X. cheopis* include integrated rodent and flea management. Insecticide application targeting flea habitats, combined with rodent control measures and housing improvements, is effective in reducing flea infestations and interrupting transmission cycles. Public health efforts must also emphasize early detection of plague and murine typhus cases, prompt treatment, and risk communication in endemic regions [13,168].

Given its central role in the epidemiology of plague and other flea-borne diseases, *X. cheopis* remains a priority target for vector surveillance and control programs, especially in areas where human and rodent populations intersect closely [168,169].

### 4.2. Ornithonyssus bacoti

*Ornithonyssus bacoti*, commonly known as the tropical rat mite, is a hematophagous ectoparasite primarily associated with commensal rodents, particularly *Rattus norvegicus* and *Rattus rattus*. Although it is not an actual flea or tick, this mite can bite humans, resulting in a condition known as rat mite dermatitis. In specific contexts, *O. bacoti* may also act as a mechanical or potential biological vector for various pathogens, underscoring its relevance in rodent-borne zoonoses (Table 2) [170,171].

Unlike fleas or lice, *O. bacoti* does not live permanently on its rodent host. Instead, it resides in nests, cracks, walls, or bedding materials, emerging periodically to feed on blood. When rodent hosts die or migrate, mites may seek alternative hosts, including humans, for blood meals. Human exposure typically occurs in urban dwellings, warehouses, or laboratories infested with rodents, and outbreaks are often associated with rodent control campaigns that displace mite populations [171,172].

Infestation by *O. bacoti* in humans causes intense pruritus, erythematous papules, and sometimes vesicular or pustular lesions, often misdiagnosed as scabies, bedbug bites, or allergic dermatitis. The dermatitis may persist for weeks and can lead to secondary bacterial infections due to scratching. Clusters of cases are often reported among residents of infested buildings or personnel working in rodent facilities [173,174].

Beyond dermatologic effects, *O. bacoti* has been experimentally implicated in the transmission of several zoonotic pathogens, including *Rickettsia akari*, *Coxiella burnetii*, *Borrelia* spp., and *Haemophilus influenzae*. While the exact epidemiological significance of these associations in natural settings remains to be fully clarified, the potential for pathogen transmission reinforces the need for surveillance and control in rodent-infested areas [175,176].

Prevalence studies have shown high infestation rates of *O. bacoti* in rodent populations from urban environments in North America, Asia, and parts of Europe. In some cities, more than 50% of captured rats harbor the mite, especially in poorly maintained structures with abundant rodent nesting sites. The mite’s resilience and ability to survive without feeding for extended periods further complicate eradication efforts [170,177].

Effective management of *O. bacoti* infestations requires an integrated approach, including rodent eradication, environmental decontamination, and application of acaricides in infested areas. Importantly, rodent removal must be accompanied by concurrent mite control to prevent the dispersal of mites and minimize the risk of human bites. Education and training of public health workers, pest control professionals, and clinicians are essential to recognize and manage mite infestations and implement appropriate interventions [170,178].

As urban rodent populations expand and infestations increase in frequency, *O. bacoti* represents a growing ectoparasitic concern with potential dermatological and vector-borne implications for human health [177].

### 4.3. Ixodes spp.

*Ixodes* spp. are hard ticks of significant medical and veterinary importance, serving as vectors for numerous zoonotic pathogens. Several species, including *Ixodes scapularis*, *Ixodes ricinus*, and *Ixodes persulcatus*, parasitize rodents during their immature stages, playing a crucial role in the transmission of bacterial, viral, and protozoal pathogens to humans and other animals (Table 2). Rodents serve as essential hosts for the larval and nymphal stages of these ticks, contributing to the maintenance and amplification of vector-borne disease cycles [179,180].

Among the most prominent diseases associated with *Ixodes* spp. is Lyme borreliosis, caused by *Borrelia burgdorferi* sensu lato. Rodents, particularly species such as *Peromyscus leucopus* (white-footed mouse) and *Apodemus flavicollis*, act as competent reservoirs for *Borrelia*, infecting feeding ticks that later transmit the pathogen to larger vertebrates, including humans. Additionally, *Ixodes* ticks transmit other pathogens such as *Anaplasma phagocytophilum* (human granulocytic anaplasmosis), *Babesia microti* (babesiosis), Tick-borne encephalitis virus (TBEV), and *Borrelia miyamotoi* [181,182,183,184,185].

The geographic distribution of *Ixodes* spp. is expanding globally, driven by climate change, altered land use, and changes in host population dynamics. As temperature and humidity influence tick survival and development, warming climates have facilitated the northward and altitudinal expansion of key *Ixodes* species in North America, Europe, and Asia. In parallel, growing rodent populations in peri-urban environments have increased opportunities for human exposure to infected ticks [183,184].

Surveillance studies consistently identify rodents as critical amplifiers of tick populations and associated pathogens. High infestation rates of larvae and nymphs are frequently observed on wild rodent species, and the overlap of rodent habitats with human recreational or residential areas increases the risk of spillover. For example, studies in temperate woodlands in Europe and North America have documented co-infection of rodents with multiple pathogens and simultaneous infestation by *Ixodes* spp., highlighting their role in the ecology of tick-borne disease complexes [186,187].

Public health strategies to reduce the risk of tick-borne diseases must consider the pivotal role of rodents. Integrated approaches involving tick control, habitat management, personal protective measures (e.g., repellents and protective clothing), and public education are essential. Rodent population control in endemic areas, combined with environmental modifications such as vegetation management, can reduce tick abundance and interrupt transmission cycles [188,189].

Given the diversity of *Ixodes*-borne pathogens and the increasing incidence of tick-borne diseases, ongoing monitoring of rodent–tick–pathogen interactions is vital. Understanding the ecological drivers of these complex systems is key to predicting and mitigating future public health threats posed by *Ixodes* spp. [190,191].

## 5. Climate Change and Anthropogenic Activities Leading to the Emergence of Rodent-Borne Zoonoses

Climate change and anthropogenic activities are key drivers in the emergence and spread of rodent-borne zoonoses. Rising global temperatures, altered precipitation patterns, and habitat disruptions are influencing rodent population dynamics, leading to increased human exposure to zoonotic pathogens. Warmer climates can expand the geographical range of rodent species [192,193,194,195], enabling them to thrive in new environments and increasing the risk of pathogen spillover. Additionally, changes in seasonal rainfall can impact food availability, driving rodents into urban and peri-urban areas where contact with humans becomes more frequent, facilitating disease transmission [5,196,197].

Anthropogenic activities such as deforestation, agricultural expansion, and urbanization exacerbate the risks associated with rodent-borne zoonoses. Habitat destruction forces rodents to migrate, often bringing them into closer proximity to human settlements, livestock, and domestic animals. This increases the likelihood of disease outbreaks, particularly for zoonotic parasites such as *Toxoplasma gondii*, *Leishmania* spp., and *Trypanosoma cruzi*. Intensive farming and irrigation projects create favourable conditions for rodent proliferation, leading to higher parasite transmission rates [5,198,199].

Climate-induced shifts in vector populations, such as fleas, ticks, and mites, further compound the risk by altering transmission cycles. Integrated surveillance, environmental management, and rodent control strategies are essential in mitigating the impact of climate change and human activities on the emergence of rodent-borne zoonotic diseases [200,201,202].

### 5.1. Habitat Modification and Behavioral Shifts in Rodents

Climate change is significantly reshaping rodent habitats and behaviours, increasing the risk of zoonotic disease transmission. Rising temperatures and shifting precipitation patterns modify ecosystems, forcing rodents to adapt by migrating to new areas, including human settlements. Changes in vegetation cycles affect food resource availability, often triggering population booms in rodent communities. Increased densities intensify intraspecies competition and may lead to heightened aggression, fostering the transmission of pathogens within and across rodent populations. These ecological alterations are closely linked to the transmission of rodent-borne zoonoses, including hantavirus infections, leptospirosis, and *Toxoplasma gondii*, underscoring the urgency of sustained ecological surveillance and disease control programs [199,203,204].

### 5.2. Impact of Extreme Weather Events

Extreme weather events, such as hurricanes, floods, droughts, and wildfires, significantly influence the emergence of rodent-borne zoonotic diseases [205,206,207]. Flooding can displace rodent populations, forcing them into human-inhabited areas, thereby increasing direct contact and contamination of water and food sources with pathogens such as *Leptospira* spp., which significantly raises the risk of leptospirosis outbreaks. Droughts reduce natural food availability, driving rodents to forage in urban environments and heightening the risk of disease spillover. Wildfires destroy habitats, prompting rodent migration and altering predator-prey dynamics, potentially increasing rodent densities [208]. These disturbances exacerbate the transmission of zoonotic pathogens, underscoring the importance of proactive disease surveillance and environmental management in disaster-prone regions [209,210,211].

### 5.3. Alteration of Pathogen Dynamics

Climate change alters pathogen dynamics by influencing the survival, replication, and transmission of zoonotic agents carried by rodents. Rising temperatures and humidity levels can enhance the persistence of bacterial, viral, and parasitic pathogens in the environment, increasing their transmission potential. Warmer conditions may also accelerate the life cycles of vectors, such as fleas and ticks, facilitating the spread of diseases like plague and Lyme disease. Additionally, altered rodent immunity and stress from environmental changes can lead to increased pathogen shedding rates. These shifts contribute to the emergence and re-emergence of rodent-borne zoonoses, necessitating improved monitoring and adaptive public health interventions [201,212,213].

## 6. Public Health Implications and Mitigation Strategies

Rodent-borne zoonotic diseases pose significant public health challenges due to their ability to spread rapidly in human populations, often leading to outbreaks with severe consequences. These diseases, including hantavirus, leptospirosis, toxoplasmosis, and plague, can cause severe morbidity and mortality, particularly in vulnerable communities with poor sanitation and limited healthcare access. The increasing interaction between rodents and human populations, driven by urbanization, climate change, and habitat destruction, further exacerbates disease risks. Additionally, rodents serve as reservoirs for emerging pathogens, making them a persistent threat in both rural and urban environments. Public health responses must address not only immediate outbreaks but also the underlying environmental and ecological factors that drive transmission [214,215,216].

Mitigation strategies require an integrated, multidisciplinary approach that includes surveillance, rodent control, and public awareness. Improved sanitation, waste management, and secure food storage help reduce rodent access to human settlements. Vector control programs targeting fleas, ticks, and mites also minimize transmission risks. Early detection systems and research on rodent–pathogen interactions are essential for predicting and managing outbreaks. Additionally, educating communities on hygiene practices and safe food handling can reduce exposure to rodent-borne pathogens. Collaboration among governments, researchers, and health agencies is crucial for developing effective policies to prevent and mitigate the impact of rodent-borne zoonotic diseases [172,201,217].

### 6.1. Surveillance and Early Warning System

Surveillance and early warning systems are essential for the timely detection and control of rodent-borne zoonotic diseases. Continuous monitoring of rodent populations, their pathogens, and environmental factors enables health authorities to identify emerging risks before outbreaks occur. Molecular diagnostics, serological testing, and ecological surveillance help track the circulation of pathogens in rodent reservoirs. Early detection allows for rapid intervention, such as targeted rodent control, vaccination campaigns, and public health alerts. Integrating these systems with climate and environmental data enhances predictive modelling, enabling proactive measures. Strengthening global surveillance networks improves preparedness, reducing the public health and economic burden of rodent-borne zoonotic diseases [218,219,220].

### 6.2. Rodent Control and Habitat Management

Rodent control and habitat management are crucial strategies for reducing the risk of rodent-borne zoonotic diseases. Effective rodent control involves monitoring the population, using traps and rodenticides, and implementing biological control measures to minimize infestations. Habitat management focuses on reducing food and shelter availability for rodents by improving waste disposal, securing food storage, and eliminating breeding sites. Urban planning, incorporating rodent-proof infrastructure and sustainable land-use practices, can further limit human-rodent interactions. Integrating these strategies with public awareness campaigns ensures long-term success in controlling rodent populations and reducing the spread of zoonotic pathogens, ultimately protecting public health [221,222,223].

### 6.3. Climate Change Adaptation and Mitigation

Climate change adaptation and mitigation play a crucial role in controlling rodent-borne zoonotic diseases by addressing the environmental factors that drive disease emergence and transmission. Adaptation strategies include strengthening surveillance systems, improving urban planning to minimize rodent habitats, and enhancing public health infrastructure to respond to outbreaks. Mitigation efforts focus on reducing greenhouse gas emissions, preserving ecosystems, and promoting sustainable agricultural practices to prevent habitat disruption. Integrated pest management and ecological restoration help regulate rodent populations naturally. By implementing climate-resilient health policies and environmental strategies, it ensures that communities are not only protected from current zoonotic threats but are also better prepared for future climate-related challenges [224,225,226].

## 7. Prevention and Control Strategies Based on the One Health Approach

Rodents and their associated parasites exemplify the complexity of host–pathogen interactions in zoonotic transmission. Different types of hosts play distinct roles in these ecological cycles. Intermediate hosts harbor developmental stages of parasites, such as rodents infected with larval *Echinococcus multilocularis*. Definitive hosts support the sexual reproduction of parasites, as in the case of rodents serving as final hosts for *Hymenolepis diminuta*. Reservoir hosts maintain the pathogen in nature, ensuring persistence even when human cases are absent, such as rodents in the epidemiology of *Trypanosoma cruzi* [227,228,229]. Within this category, the notion of a competent reservoir is crucial, referring to rodent species that not only sustain but also effectively transmit pathogens to vectors or humans, such as *Peromyscus leucopus* for *Borrelia burgdorferi*. Finally, some species function as essential reservoirs, without which the pathogen cannot persist in a given ecosystem, making them indispensable targets for surveillance and control. Recognizing these distinctions is fundamental for applying One Health-based interventions that are ecologically sound and epidemiologically effective [230,231,232].

The One Health approach emphasizes the interconnectedness of human, animal, and environmental health, providing a comprehensive framework for preventing and controlling rodent-borne parasitic diseases. Effective strategies require integrated surveillance systems that simultaneously monitor rodent populations, vectors, and environmental contamination to manage these issues effectively. Molecular and ecological tools should be employed to identify reservoir species, track parasite genotypes, and detect early signs of emerging zoonoses. Such systems must be linked with human health surveillance to ensure timely interventions [233,234,235].

Environmental management remains a cornerstone of prevention. Improvements in sanitation, waste disposal, and food storage reduce opportunities for rodents to thrive near human settlements. Habitat modifications that limit rodent nesting sites, combined with vector control programs targeting fleas, ticks, and mites, help interrupt multiple pathways of transmission. Climate-adaptive strategies are equally important, given that warming and altered precipitation patterns are expanding rodent ranges and intensifying interactions with humans [236,237,238].

Rodent control measures should be carefully designed to minimize ecological disruption. Mechanical traps, biological control, and integrated pest management are preferable to indiscriminate chemical rodenticides, which may harm non-target species and disrupt ecosystems. Public education campaigns can reinforce community-level hygiene practices, encourage safe food preparation, and reduce behaviors that heighten human exposure to contaminated environments [239,240,241].

Finally, multisectoral collaboration is indispensable. Veterinary, medical, and environmental experts must coordinate with policymakers and local communities to design sustainable interventions. By aligning efforts across disciplines and sectors, the One Health approach not only addresses the immediate threat of rodent-borne parasites but also builds resilience against future zoonotic challenges. In this way, prevention and control strategies can shift from reactive measures to proactive, long-term protection of human and animal health in shared ecosystems [242,243,244].

## 8. Limitations

The variability and heterogeneity of available data across geographic regions, rodent species, and diagnostic methodologies limit the scope of this review. Many studies focus on specific parasites or localities, hindering broader generalization. Additionally, underreporting and lack of standardized surveillance in resource-limited settings may lead to an underestimation of true prevalence and public health impact. The review also relies heavily on published literature, which may introduce publication bias and overlook unpublished data or grey literature relevant to rodent-borne parasitic diseases.

## 9. Conclusions

Although many of the zoonotic pathogens associated with rodents are responsible for relatively rare human infections, their importance should not be underestimated. Even rare spillover events can have significant epidemiological consequences when ecological or anthropogenic factors shift. For instance, changes in rodent population density, behavior, or habitat—often influenced by urbanization, agricultural practices, or climate variability—can increase the likelihood of human contact and pathogen transmission. In such scenarios, rodents may serve as amplifiers or maintainers of pathogens, enabling sporadic cases to evolve into localized outbreaks. Furthermore, in a globalized context where human mobility and trade can rapidly spread pathogens beyond their original ecological niches, the role of rodents becomes even more relevant for anticipating and preventing emerging infectious disease threats. Thus, understanding rodent ecology and their role in the epidemiology of both common and rare zoonoses is critical for effective surveillance and public health preparedness.

Rodent-borne parasitic diseases pose a significant and escalating threat to global public health. This review highlights the diverse protozoan, helminthic, and ectoparasitic pathogens carried by rodents that can be transmitted to humans through various ecological and environmental pathways. The interactions between rodents, their parasites, and human populations are intensifying due to urbanization, climate change, and habitat disruption. The emergence and re-emergence of diseases such as toxoplasmosis, leishmaniasis, giardiasis, cryptosporidiosis, and trichinellosis underscore the need for a One Health approach, integrating surveillance, rodent control, environmental management, and public education. Surveillance systems must be strengthened to detect zoonotic threats early, while multisectoral collaboration is essential for effective intervention. Reducing the burden of rodent-borne parasitic diseases requires proactive public health strategies, particularly in vulnerable communities. As global environmental conditions continue to shift, coordinated and sustained efforts are essential to mitigate the burden of rodent-associated parasitic infections. Comprehensive monitoring, prevention, and control strategies must be applied not only in endemic regions but also in areas where disease emergence is anticipated due to ecological change.

## Figures and Tables

**Figure 1 animals-15-02681-f001:**
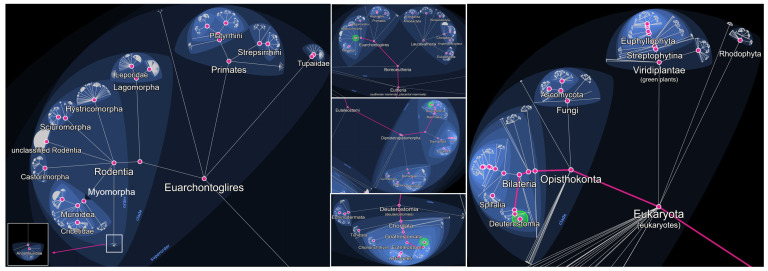
Taxonomic location of the order Rodentia and its suborders. Constructed using LifeMap, https://lifemap.cnrs.fr/, accessed on 1 August 2025.

**Figure 2 animals-15-02681-f002:**
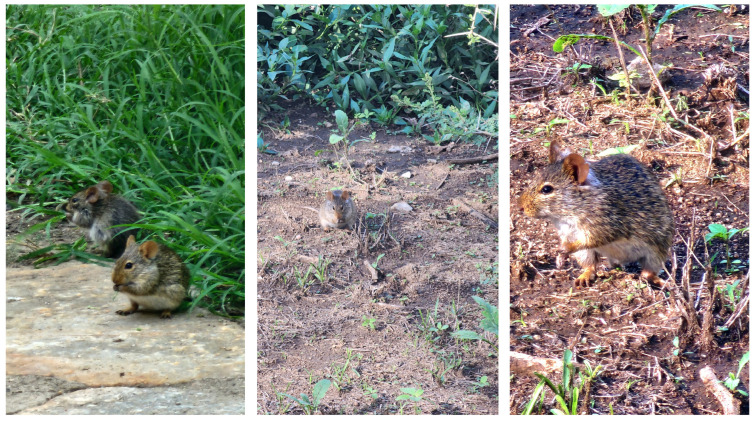
African grass rat (*Arvicanthis niloticus*). Species of rodent in the family Murinae. Photo taken by AJRM at Serengeti National Park, Tanzania, in December 2024, near humans and other wild animals.

**Figure 3 animals-15-02681-f003:**
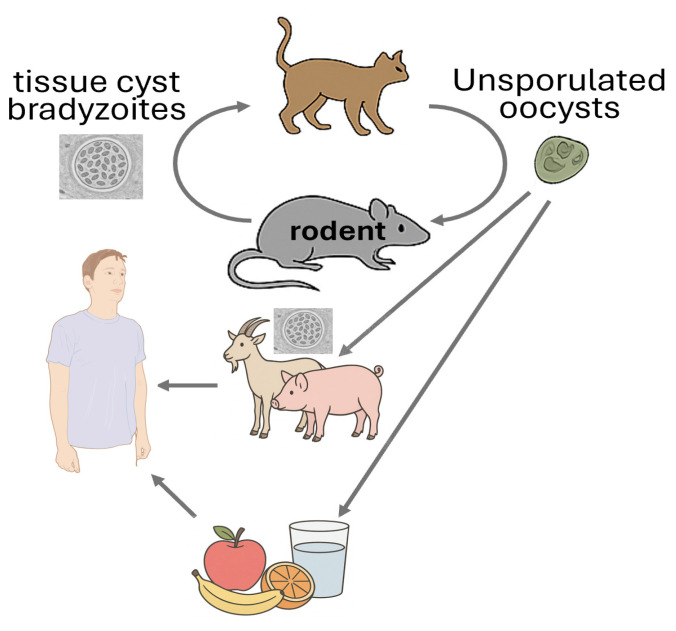
Life cycle of toxoplasmosis. *Toxoplasma gondii* infects members of the Felidae family, with domestic cats serving as the definitive hosts. They shed unsporulated oocysts in feces, which become infective after sporulating in the environment. Rodents and other animals act as intermediate hosts by ingesting contaminated soil, water, or vegetation. In these hosts, oocysts release tachyzoites that develop into tissue cysts. Cats become infected by consuming these cysts or ingesting sporulated oocysts directly. Humans acquire infection through undercooked meat, contaminated food or water, blood transfusion, organ transplant, or congenitally. Infected humans may carry tissue cysts in muscles, brain, and eyes for life. Diagnosis is mainly serological, though molecular methods like PCR can detect congenital infection.

**Figure 4 animals-15-02681-f004:**
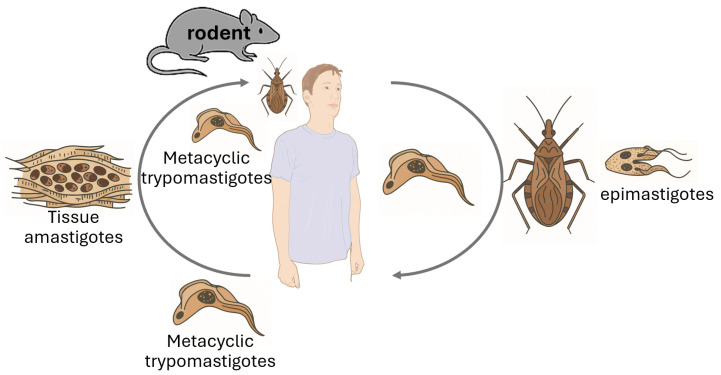
Life cycle of American trypanosomiasis. An infected triatomine transmits *Trypanosoma cruzi* by releasing trypomastigotes in its feces near the bite site. These enter the host through the wound or mucous membranes. Once inside, the parasites invade nearby cells, transform into amastigotes, and multiply by binary fission. They then convert back into trypomastigotes, which circulate in the blood and infect new cells. This cycle contributes to disease symptoms. Bloodstream trypomastigotes do not replicate and must invade cells or be ingested by another vector to continue development. When a triatomine feeds on an infected host, including rodents, it ingests the parasites, which transform into epimastigotes in its midgut, multiply, and differentiate into infective metacyclic trypomastigotes in the hindgut. Other transmission routes include blood transfusion, organ transplant, congenital infection, and consumption of contaminated food or drink.

**Figure 5 animals-15-02681-f005:**
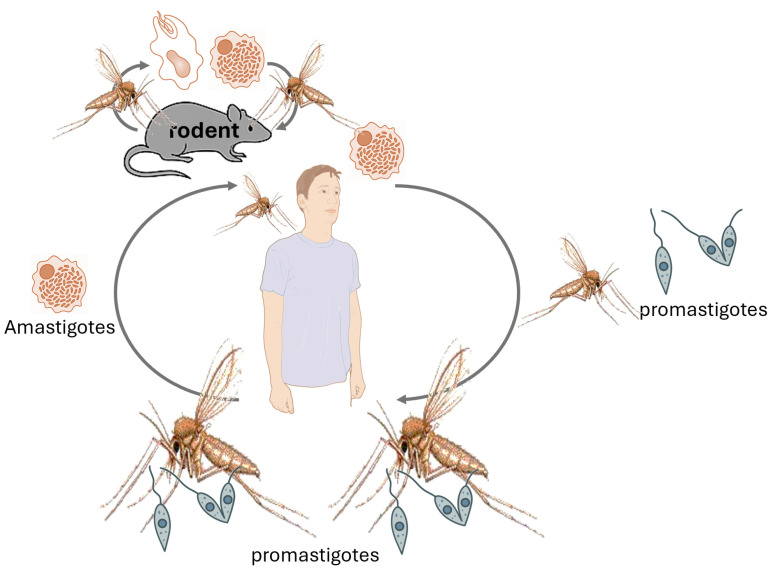
Life cycle of leishmaniasis. *Leishmania* is transmitted through the bite of infected female phlebotomine sandflies, which inject promastigotes into the skin during feeding. These promastigotes are engulfed by macrophages and other phagocytic cells, where they transform into amastigotes. Inside these cells, they multiply and spread to other mononuclear phagocytes. The progression to cutaneous or visceral disease depends on parasite and host factors. Sandflies become infected by ingesting infected cells during a blood meal. In the sandfly, amastigotes convert back into promastigotes, develop in the gut (specifically, the hindgut and midgut for Viannia, and the midgut for Leishmania), and then migrate to the proboscis to continue the cycle. Female phlebotomine sandflies may also bite different animals, transmitting *Leishmania* promastigotes, including rodents.

**Figure 6 animals-15-02681-f006:**
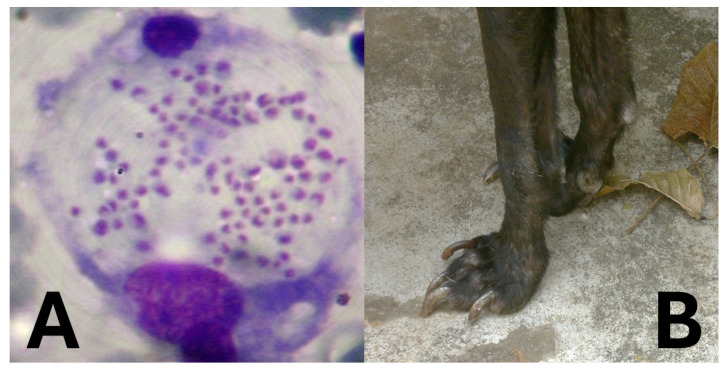
Amastigotes of *Leishmania* (**A**) and onychogryphosis in a dog with visceral leishmaniasis (**B**). Photos taken by AJRM.

**Figure 7 animals-15-02681-f007:**
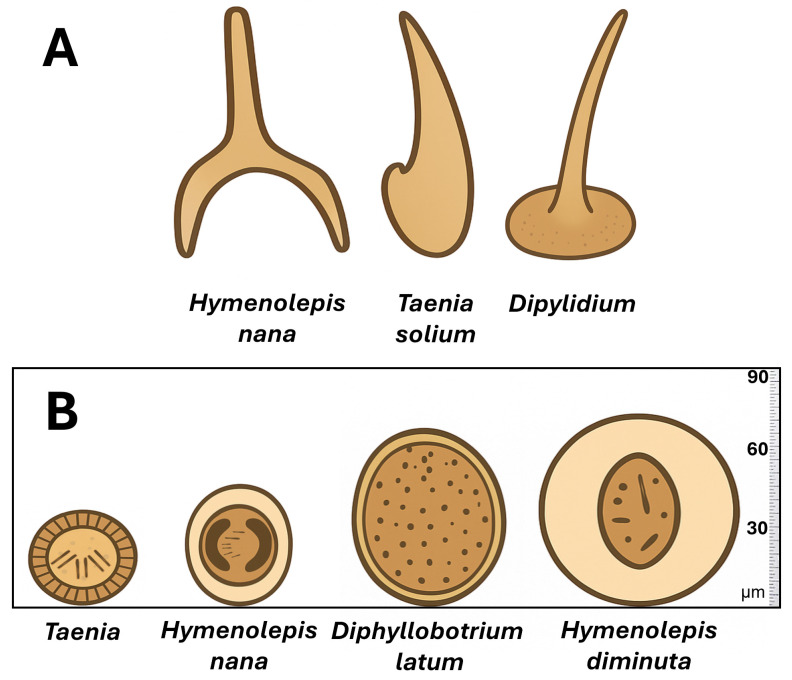
*Hymenolepis* spp. morphological characteristics. (**A**) Representations of the scolex (hooks) of *H. nana* and other cestodes. (**B**) Comparison of the aspects and size of eggs of *H. nana, H. diminuta, Taenia* and *Diphyllobotrium latum*. Based on Lamberth RA, 1969 [99].

**Table 1 animals-15-02681-t001:** Suborders of the Order Rodentia.

Suborders	Contents
Anomaluromorpha	anomalures and springhares
Castorimorpha	beavers and others
Hystricomorpha	guinea pigs and others
Myomorpha	mice and others
Sciuromorpha	squirrels

Modified from https://www.ncbi.nlm.nih.gov/Taxonomy/Browser/wwwtax.cgi, accessed on 1 August 2025.

**Table 2 animals-15-02681-t002:** Main parasites associated with rodents and their features.

Parasite	Parasitological Features	Clinical Impacts	Forms of Transmission	Preventive Measures
*Toxoplasma gondii*	Intracellular protozoan; form tissue cysts	Toxoplasmosis is severe among immunocompromised individuals and pregnancy complications	Ingestion of oocysts in contaminated food, water, or cat faeces. Ingestion of tissue cysts in undercooked or raw meat from infected animals.	Proper cooking of meat; hygiene in handling cat faeces
*Trypanosoma cruzi*	Flagellated protozoan; transmitted by triatomine bugs	Chagas disease, cardiomyopathy, and digestive disorders	Bite of an infected triatomine bug; contaminated food or transfusions	Vector control, improved housing, and screening
*Leishmania* spp.	Flagellated protozoan; transmitted by sandflies	Leishmaniasis: cutaneous, mucosal, and visceral forms	Bite of infected sandflies	Use of insect repellent; vector control
*Giardia intestinalis*	Flagellated protozoan; causes intestinal infection	Giardiasis: chronic diarrhea and malabsorption	Ingestion of cysts in contaminated water or food	Safe drinking water, proper sanitation
*Cryptosporidium* spp.	Apicomplexan parasite; waterborne transmission	Cryptosporidiosis: severe diarrhea, especially in immunocompromised patients	Ingestion of oocysts in contaminated water or direct contact	Water treatment and sanitation
*Hymenolepis diminuta*	Cestode; requires an intermediate arthropod host	Mild gastrointestinal symptoms	Ingestion of infected arthropods	Avoiding ingestion of contaminated arthropods
*Hymenolepis nana*	Cestode; directly infective to humans	Gastrointestinal discomfort and autoinfection	Ingestion of eggs in contaminated food or autoinfection	Proper hygiene and sanitation
*Trichinella spiralis*	Nematode; infects muscle tissue	Trichinellosis: muscle pain, fever, and organ damage	Consumption of undercooked infected meat	Cooking meat thoroughly
*Angiostrongylus cantonensis*	Nematode; affects CNS	Eosinophilic meningitis; neurological symptoms	Ingestion of larvae in raw or undercooked snails and slugs	Avoiding raw snails/slugs; proper food handling
*Capillaria hepatica*	Nematode; liver infection	Hepatic capillariasis; liver dysfunction	Ingestion of eggs from contaminated soil or food	Avoiding consumption of contaminated food
*Baylisascaris procyonis*	Nematode; severe neurotropic potential	Neural larva migrans; severe neurological damage	Accidental ingestion of eggs from infected raccoons or rodents	Rodent control: avoiding contaminated environments
*Xenopsylla cheopis*	Flea: vector of *Yersinia pestis*	Plague: bubonic, septicemic, and pneumonic forms	Bite of infected fleas	Flea control; rodent management
*Ornithonyssus bacoti*	Mite: causes dermatitis	Mite dermatitis, skin irritation, and secondary infections	Direct contact with rodents or contaminated environments	Rodent control; personal protection
*Ixodes* spp.	Tick: transmits bacterial and viral infections	Lyme disease and other tick-borne illnesses	Bite of infected ticks	Use of tick repellents; habitat control

**Table 3 animals-15-02681-t003:** Top ten countries with the highest numbers of visceral leishmaniasis cases reported in 2023, according to the World Health Organization. (https://www.who.int/data/gho/data/indicators/indicator-details/GHO/number-of-cases-of-visceral-leishmaniasis-reported, accessed on 1 August 2025).

Country	Cases	Population Estimates *	Incidence Rate **
Sudan	3571	50,040,000	7.14
Ethiopia	1482	128,690,000	1.15
Brazil	1461	211,140,000	0.69
Kenya	1252	55,340,000	2.26
South Sudan	778	11,480,000	6.78
Somalia	712	18,360,000	3.88
India	538	1,438,000,000	0.04
Eritrea	376	3,470,000	10.84
Yemen	240	39,390,000	0.61
Uganda	195	48,660,000	0.40

* For 2023, https://www.worldometers.info/world-population/india-population/, accessed on 1 August 2025. ** Cases per 100,000 pop.

**Table 4 animals-15-02681-t004:** Confirmed trichinellosis cases and rates per 100,000 population by country and year, EU/EEA, 2018–2022, according to the ECDC (https://www.ecdc.europa.eu/sites/default/files/documents/trichinellosis-annual-epidemiological-report-2022.pdf?utm_source=chatgpt.com, accessed on 1 August 2025).

Country	2018		2019		2020		2021		2022	
	Number	Rate	Number	Rate	Number	Rate	Number	Rate	Number	Rate
Austria	2	0.02	1	0.01	6	0.07	10	0.11	2	0.02
Belgium	0	NRC	NDR	NRC	NDR	NRC	0	NRC	0	NRC
Bulgaria	45	0.64	55	0.79	13	0.19	29	0.42	9	0.13
Croatia	0	0.00	3	0.07	0	0.00	17	0.42	NDR	NRC
Cyprus	0	0.00	0	0.00	0	0.00	0	0.00	0	0.00
Czechia	0	0.00	0	0.00	0	0.00	0	0.00	0	0.00
Denmark	NDR	NRC	NDR	NRC	NDR	NRC	NDR	NRC	NDR	NRC
Estonia	0	0.00	0	0.00	0	0.00	0	0.00	1	0.08
Finland	0	0.00	0	0.00	0	0.00	0	0.00	0	0.00
France	0	0.00	2	0.00	1	0.00	2	0.00	15	0.02
Germany	0	0.00	3	0.00	1	0.00	2	0.00	0	0.00
Greece	0	0.00	0	0.00	0	0.00	0	0.00	0	0.00
Hungary	2	0.02	0	0.00	0	0.00	0	0.00	0	0.00
Iceland	0	0.00	0	0.00	0	0.00	0	0.00	0	0.00
Ireland	0	0.00	0	0.00	0	0.00	0	0.00	0	0.00
Italy	2	0.00	10	0.02	79	0.13	0	0.00	4	0.01
Latvia	1	0.05	1	0.05	1	0.05	7	0.37	3	0.16
Liechtenstein	NDR	NRC	NDR	NRC	NDR	NRC	0	0.00	0	0.00
Lithuania	0	0.00	0	0.00	0	0.00	1	0.04	0	0.00
Luxembourg	0	0.00	0	0.00	0	0.00	0	0.00	0	0.00
Malta	0	0.00	0	0.00	0	0.00	0	0.00	0	0.00
Netherlands	0	0.00	1	0.01	0	0.00	0	0.00	0	0.00
Norway	0	0.00	0	0.00	0	0.00	0	0.00	0	0.00
Poland	2	0.01	2	0.01	11	0.03	2	0.01	1	0.00
Portugal	0	0.00	1	0.01	0	0.00	0	0.00	0	0.00
Romania	10	0.05	6	0.03	4	0.02	6	0.03	4	0.02
Slovakia	0	0.00	0	0.00	0	0.00	0	0.00	0	0.00
Slovenia	0	0.00	0	0.00	0	0.00	0	0.00	0	0.00
Spain	2	0.00	12	0.03	1	NRC	1	NRC	0	0.00
Sweden	0	0.00	0	0.00	0	0.00	0	0.00	0	0.00
**EU/EEA** **(30 countries)**	**66**	**0.02**	**97**	**0.02**	**117**	**0.03**	**77**	**0.02**	**39**	**0.01**
United Kingdom	0	0.00	0	0.00	NA	NA	NA	NA	NA	NA
**EU/EEA** **(31 countries)**	**66**	**0.01**	**97**	**0.02**	**117**	**0.03**	**NA**	**NA**	**NA**	**NA**

Source: Country reports. NA: not applicable; NDR: no data reported; NRC: no rate calculated. No data from 2020 onwards were reported by the United Kingdom, due to its withdrawal from the EU on 31 January 2020. EU/EEA, European Union/European Economic Area.

## Data Availability

Not applicable.

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
