# Peer review of "Rodent-Borne Parasites and Human Disease: A Growing Public Health Concern"

_animals, 2025, doi:10.3390/ani15182681_

Round 1
Reviewer 1 Report
Comments and Suggestions for Authors
Line 92: in Italics
Table 1. - Toxoplasma gondii - Form of transmission - tissue cyst in animals ??
Table 1. - Xenopsylla cheopis - Yersinia pestis - in Italics
Line 124: in Italics
Line 558 and 559: in Italics
Line 608: Leishmania spp. ?
Author Response
Reviewer 1
Line 92: in Italics
Response: Done. Corrected.
Table 1. - Toxoplasma gondii - Form of transmission - tissue cyst in animals ??
Response: Done. Included in the Table.
Table 1. - Xenopsylla cheopis - Yersinia pestis - in Italics
Response: Done. Corrected.
Line 124: in Italics
Response: Done. Corrected.
Line 558 and 559: in Italics
Response: Done. Corrected.
Line 608: Leishmania spp. ?
Response: Done. Corrected.
Reviewer 2 Report
Comments and Suggestions for Authors
General comments:
The topic of the review is relevant; to my knowledge, the last similar work was published in 2019 (Mustapha et al.; SAJP, 3(3): 1-15, Article no.SAJP.53929) and did not include ectoparasites. The manuscript is written in good English, well-structured, with conclusions supported by the listed citations. Most of the cited references are not older than 5 years, and self-citation is reasonable (15 papers by the first author out of a total of 180). Regarding specific content, I consider the selection of protozoa, helminths and ectoparasites to be adequate, but I recommend also including the tapeworm Echinococcus multilocularis, the causative agent of alveolar echinococcosis, one of the most pathogenic zoonoses in Europe, with a current increase in the number of cases. Also, Babesia microti (causative agent of tick-borne human babesiosis) and Entamoeba histolytica (causative agent of amoebiasis) could be at least briefly mentioned. In terms of the formal aspects of the work, I consider the tables to be well prepared and easy to understand, but the numbering of the figures is incorrect, and I have factual comments on some of the figures, which are listed below.
Specific comments:
- Lines 49-51 and Figure 1: The group Dermaptera is missing from the description of the order Euarchontoglires (see, for example, Esselstyn et al., 2017, DOI:10.1093/gbe/evx168).
- Lines 51 – 54 and Figure 1: The illustration of suborders of the order Rodentia should reflect the monophyly of the mouse-related clade (the suborder Supramyomorpha), covering infraorders Anomaluromorphi, Castorimorphi and Myomorphi (see D´Elía et al., 2019, DOI:10.1093/jmammal/gyy179).
- In Figure 1, showing the taxonomic location of rodents within all eukaryotes is redundant; moreover, the middle parts have poor resolution and are unclear. It would be more informative to elaborate better the classification of the Euarchontoglires group and the order Rodentia, i.e., to enlarge and adjust the left part of the figure and omit the rest.
- Figure 2 would be more illustrative if it showed representatives of the three main groups of rodents (Sciuromorpha, Supramyomorpha, Hystricomorpha), rather than three different photos of the same species.
- Line 79: The Leishmania genus is transmitted by insect vectors (sand flies) and cannot be mentioned among pathogens transmitted by contaminated water.
- Line 94: Although lice transmit the causative agent of relapsing fever, they are species-specific and do not have zoonotic potential, so I recommend not mentioning them here.
- Table 1: T. cruzi and G. intestinalis are characterised as “flagellated protozoan”; the same should apply to Leishmania. In Ixodes spp. description, not only bacterial but also viral infections should be mentioned.
- In Chapter 2.1. on Toxoplasma gondii, it could be mentioned that this parasite was first described in a rodent, Ctenodactylus gundi (Nicolle and Manceaux, 1909). Modification of the behaviour of infected humans can also be stated (see DOI: 10.1007/978-3-662-43978-4_3464).
- Line 158: It should be noted here that the methodology of the above-cited articles did not allow for differentiation between T. cruzi and the non-pathogenic T. rangeli, so these data on the prevalence of T. cruzi in rodents may be overestimated.
- Line 166: The references cited after the introductory sentence (31–33) focus on individual countries or regions of South and Central America, but a more general review should be cited, covering the diversity of Leishmania species, including those from the Old World.
- Figure 5: It is confusing that promastigotes are depicted beneath the image of a human, as this form is only present in phlebotomine sand flies. Also, the closed cycle around the rodent does not make sense, as it suggests that the parasite is transmitted between rodents without the involvement of an insect vector.
- Line 171: “especially dogs” is true for infantum, not generally for the genus Leishmania
- Line 191: Leishmania is transmitted
- Line 197: hindgut and midgut for Viannia
- Line 203: “This high infection rate suggests that urban rodent populations can serve as significant reservoirs for the parasite” High prevalence is not a sufficient evidence to incriminate a reservoir host of Leishmania (see World Health Organization. Control of the leishmaniases. WHO Technical Report Series no. 949. Geneva: The Organization; 2010; Chaves et al. 2007, DOI: 10.1016/j.pt.2007.05.003). Xenodiagnostic experiments are needed to confirm the role of the suspected host species in Leishmania transmission. Authors should add this information and can provide data on Leishmania-rodent pairs, where the reservoir role of the rodent species has been confirmed by xenodiagnosis. For example, infantum – Rattus rattus (Gradoni et al 1983, Trans R Soc Trop Med Hyg 77:427-431), L. panamensis - Proechimys semispinosus (Travi et al. 2002, DOI: 10.1590/s0074-02762002000600025), L. braziliensis - Necromys lasiurus, Nectomys squamipesand Rattus rattus, Andrade et al. 2015 DOI: 10.1371/journal.pntd.0004137, L. major – Meriones shawi (Sadlova et al. 2023, DOI: 10.3390/pathogens12040614).
- Lines 257-259: The general characterisation of the genus Cryptosporidium should refer to a general review (for example, DOI: 1016/S0020-7519(00)00135-1), not to specific papers describing cases in one country (54,55).
- Line 296: The typical manifestation of leishmaniasis in dogs is skin lesions rather than onychogryphosis, so it would be good to supplement or replace the image.
- Lines 384-385: The authors cite the first finding of A. cantonensis in rats in Spain, so it might be worth mentioning here that this parasite has also been reported from continental Europe.
- Lines 599-600: Is the assumption that ‘Warmer climates can expand the geographical range of rodent species’ supported by any published study? If so, it should be cited.
Typos:
Line 92: Italicise Rickettsia typhi
Line 124: Italicise T. gondii
Line 106: Figure 3
Line 115: Is the link to Table 1 correct?
Line 133: Figure 4
Lines 169, 170: Figure 5
Line 171: Figure 6
Line 232 Italicise Giardia intestinalis
Line 293: Figure 7
Lines 351-352: italicise T. spiralis
Lines 558-9: italicise Ixodes scapularis, Ixodes ricinus, Ixodes persulcatus
Author Response
Reviewer 2
General comments:
The topic of the review is relevant; to my knowledge, the last similar work was published in 2019 (Mustapha et al.; SAJP, 3(3): 1-15, Article no.SAJP.53929) and did not include ectoparasites. The manuscript is written in good English, well-structured, with conclusions supported by the listed citations. Most of the cited references are not older than 5 years, and self-citation is reasonable (15 papers by the first author out of a total of 180).
Response: Thanks for your good comments.
Regarding specific content, I consider the selection of protozoa, helminths and ectoparasites to be adequate, but I recommend also including the tapeworm Echinococcus multilocularis, the causative agent of alveolar echinococcosis, one of the most pathogenic zoonoses in Europe, with a current increase in the number of cases.
Response: Done. Now included. We developed a new subsection for E. multilocularis.
Also, Babesia microti (causative agent of tick-borne human babesiosis) and Entamoeba histolytica (causative agent of amoebiasis) could be at least briefly mentioned.
Response: Done. Now included. We developed two new subsections for Babesia microti and Entamoeba histolytica.
In terms of the formal aspects of the work, I consider the tables to be well prepared and easy to understand
Response: Thanks for your comments.
but the numbering of the figures is incorrect,
Response: Revised and corrected.
and I have factual comments on some of the figures, which are listed below.
Specific comments:
- Lines 49-51 and Figure 1: The group Dermaptera is missing from the description of the order Euarchontoglires (see, for example, Esselstyn et al., 2017, DOI:10.1093/gbe/evx168).
Response: Euarchontoglires is not an order; it is a superorder. The Rodentia is an order, and is highlighted in Figure 1. Dermaptera is an order of arthropods. You probably mean Dermoptera, which is another order in the superorder Euarchontoglires. The Figure is focused just on the order Rodentia.
- Lines 51 – 54 and Figure 1: The illustration of suborders of the order Rodentia should reflect the monophyly of the mouse-related clade (the suborder Supramyomorpha), covering infraorders Anomaluromorphi, Castorimorphi and Myomorphi (see D´Elía et al., 2019, DOI:10.1093/jmammal/gyy179).
Response: Figure 1 is not a proper phylogenetic analysis. And again, it is just to show the location of the order Rodentia.
- In Figure 1, showing the taxonomic location of rodents within all eukaryotes is redundant; moreover, the middle parts have poor resolution and are unclear. It would be more informative to elaborate better the classification of the Euarchontoglires group and the order Rodentia, i.e., to enlarge and adjust the left part of the figure and omit the rest.
Response: We have added a table showing the suborders of the order Rodentia.
- Figure 2 would be more illustrative if it showed representatives of the three main groups of rodents (Sciuromorpha, Supramyomorpha, Hystricomorpha), rather than three different photos of the same species.
Response: Figure 2 is just illustrative.
- Line 79: The Leishmania genus is transmitted by insect vectors (sand flies) and cannot be mentioned among pathogens transmitted by contaminated water.
Response: Corrected.
- Line 94: Although lice transmit the causative agent of relapsing fever, they are species-specific and do not have zoonotic potential, so I recommend not mentioning them here.
Response: Done. Corrected.
- Table 1: T. cruzi and G. intestinalis are characterised as “flagellated protozoan”; the same should apply to Leishmania. In Ixodes spp. description, not only bacterial but also viral infections should be mentioned.
Response: Done. Both things corrected.
- In Chapter 2.1. on Toxoplasma gondii, it could be mentioned that this parasite was first described in a rodent, Ctenodactylus gundi (Nicolle and Manceaux, 1909). Modification of the behaviour of infected humans can also be stated (see DOI: 10.1007/978-3-662-43978-4_3464).
Response: Done included, both.
- Line 158: It should be noted here that the methodology of the above-cited articles did not allow for differentiation between T. cruzi and the non-pathogenic T. rangeli, so these data on the prevalence of T. cruzi in rodents may be overestimated.
Response: We included a comment about it.
- Line 166: The references cited after the introductory sentence (31–33) focus on individual countries or regions of South and Central America, but a more general review should be cited, covering the diversity of Leishmania species, including those from the Old World.
Response: Some references from the Old World leishmaniasis were also included.
- Figure 5: It is confusing that promastigotes are depicted beneath the image of a human, as this form is only present in phlebotomine sand flies. Also, the closed cycle around the rodent does not make sense, as it suggests that the parasite is transmitted between rodents without the involvement of an insect vector.
Response: Both things were improved and clarified now in the Figure.
- Line 171: “especially dogs” is true for infantum, not generally for the genus Leishmania
Response: Done. Corrected.
- Line 191: Leishmania is transmitted
Response: Done. Corrected.
- Line 197: hindgut and midgut for Viannia
Response: Done. Corrected.
- Line 203: “This high infection rate suggests that urban rodent populations can serve as significant reservoirs for the parasite” High prevalence is not a sufficient evidence to incriminate a reservoir host of Leishmania (see World Health Organization. Control of the leishmaniases. WHO Technical Report Series no. 949. Geneva: The Organization; 2010; Chaves et al. 2007, DOI: 10.1016/j.pt.2007.05.003). Xenodiagnostic experiments are needed to confirm the role of the suspected host species in Leishmania transmission. Authors should add this information and can provide data on Leishmania-rodent pairs, where the reservoir role of the rodent species has been confirmed by xenodiagnosis. For example, infantum – Rattus rattus (Gradoni et al 1983, Trans R Soc Trop Med Hyg 77:427-431), L. panamensis - Proechimys semispinosus (Travi et al. 2002, DOI: 10.1590/s0074-02762002000600025), L. braziliensis - Necromys lasiurus, Nectomys squamipesand Rattus rattus, Andrade et al. 2015 DOI: 10.1371/journal.pntd.0004137, L. major – Meriones shawi (Sadlova et al. 2023, DOI: 10.3390/pathogens12040614).
Response: Thanks. We include all such considerations in the text.
- Lines 257-259: The general characterisation of the genus Cryptosporidium should refer to a general review (for example, DOI: 1016/S0020-7519(00)00135-1), not to specific papers describing cases in one country (54,55).
Response: We included now that suggested reference and another general from Clinical Microbiology Reviews.
- Line 296: The typical manifestation of leishmaniasis in dogs is skin lesions rather than onychogryphosis, so it would be good to supplement or replace the image.
Response: Both are manifestations of leishmaniasis in dogs. So, the image is ok.
- Lines 384-385: The authors cite the first finding of A. cantonensis in rats in Spain, so it might be worth mentioning here that this parasite has also been reported from continental Europe.
Response: Yes, now we included that.
- Lines 599-600: Is the assumption that ‘Warmer climates can expand the geographical range of rodent species’ supported by any published study? If so, it should be cited.
Response: Done. We cited some related studies.
Typos:
Line 92: Italicise Rickettsia typhi
Response: Done.
Line 124: Italicise T. gondii
Response: Done.
Line 106: Figure 3
Response: Done, corrected.
Line 115: Is the link to Table 1 correct?
Response: All were corrected now.
Line 133: Figure 4
Response: Done, corrected.
Lines 169, 170: Figure 5
Response: Done, corrected.
Line 171: Figure 6
Response: Done, corrected.
Line 232 Italicise Giardia intestinalis
Response: Done, corrected.
Line 293: Figure 7
Response: Done, corrected.
Lines 351-352: italicise T. spiralis
Response: Done, corrected.
Lines 558-9: italicise Ixodes scapularis, Ixodes ricinus, Ixodes persulcatus
Response: Done, corrected.
Reviewer 3 Report
Comments and Suggestions for Authors
Dear authors. Please find mi revision in the corresponding file

The use of language is good, the manuscript does not need to be improved. Just check some misspelled marked in the revised fil
Author Response
Reviewer 3
The use of language is good, the manuscript does not need to be improved. Just check some misspelled marked in the revised fil
Response: Thanks for your comments.
Dear authors, below you will find my main recommendations on your manuscript. Details of specifications of my revision are marked in the file.
Response: Thanks
This manuscript is an excellent opportunity to review the concepts of the different types and functions of hosts in zoonotic pathogen transmission (for example intermediate host, reservoir host, definitive host, competent reservoir, essential reservoir). For parasitologists, particularly those of the old school, the roles of the different hosts participating in the biological cycle of parasitic pathogens, as well as their epidemiological role in transmission, are probably sufficiently clear, but this is not the case for researchers without parasitological training. This manuscript highlights some host terms whose definition does not correspond to the context of the paragraph. I suggest authors include a section for this purpose focusing on the role of rodents.
Response: Thanks. We have included that now, at the Introduction.
I made throughout the file general comments as:
– Clarify statements and ideas in some paragraphs.
– Include some missing references.
– Check wording.
Response: Done. Now has been improved.
To avoid being repetitive about disease prevention and control measures, I strongly recommend creating a section outlining prevention and control strategies based on the One Health approach, specifying if necessary a pathogen, geography, or transmission scenario. I believe this would be more educational than doing so separately by pathogen. Furthermore, it would be very informative to mention successful cases in the context of prevention of zoonotic diseases with rodents as hosts and based on the One Health approach. Additionally, I recommend emphasizing implementation of sustainable strategies and not promoting the use of chemicals as the sole control strategy.
Response: We have improved the document incorporating more about One Health.
Most of the zoonotic pathogens listed in the manuscript are causes of rare human diseases, so explain the relevance of the rodents in the epidemiology of those diseases to make this importance clear. For example, pointing out that in rare cases of transmission to humans, epidemiological factors related to rodents could change the epidemiological...
Response: We have now included a new paragraph about it in the Conclusions.
Round 2
Reviewer 3 Report
Comments and Suggestions for Authors
Dear authors. I revised your improved version of the manuscript and I found you attended most of the suggestions made on your first version. However, there are some aspects you decided not to change or eliminate, for example the illustrations of morphological characteristics of Hymenolepis. You did not include a section outlining prevention and control strategies based on the One Health approach, nor the concepts of the different types and functions of hosts in zoonotic pathogen transmission foci (for example, intermediate host, reservoir host, definitive host, competent reservoir, essential reservoir) were not sufficiently analyzed. I will let the editor to make the final decision on your manuscript.
Comments on the Quality of English LanguageNo comments
Author Response
Reviewer 3 – Comments and Responses
Dear authors. I revised your improved version of the manuscript and I found you attended most of the suggestions made on your first version. However, there are some aspects you decided not to change or eliminate, for example the illustrations of morphological characteristics of Hymenolepis.
Response: Thanks. We carefully checked on the figure 7, and we consider it useful especially for those not familiar with the diagnosis of hymenolepiasis, then we did not delete this image. Additionally, we did not see an specific comment previously asking for that, and more, the reason to delete it.
You did not include a section outlining prevention and control strategies based on the One Health approach, nor the concepts of the different types and functions of hosts in zoonotic pathogen transmission foci (for example, intermediate host, reservoir host, definitive host, competent reservoir, essential reservoir) were not sufficiently analyzed. I will let the editor to make the final decision on your manuscript.
Response: Thanks. We have now included in the latest version, and new section, 7, about it, 7. Prevention and Control Strategies Based on the One Health Approach. Wew also took the opportunity at thew starting of this section to include the concepts of the different types and functions of hosts in zoonotic pathogen transmission foci (for example, intermediate host, reservoir host, definitive host, competent reservoir, essential reservoir).